# Evidence of the Importance of Dietary Habits Regarding Depressive Symptoms and Depression

**DOI:** 10.3390/ijerph17051616

**Published:** 2020-03-02

**Authors:** Tina Ljungberg, Emma Bondza, Connie Lethin

**Affiliations:** 1The Nurse Unit in the Municipality of Trelleborg, 231 83 Trelleborg, Sweden; tina.ljungberg@trelleborg.se; 2The Nurse Unit in the Municipality of Eslöv, 241 80 Eslöv, Sweden; emma.bondza@eslov.se; 3Department of Health Sciences, Faculty of Medicine, Lund University, Box 157, SE-221 00 Lund, Sweden; 4Faculty of Medicine, Department of Clinical Sciences, Clinical Memory Research Unit, Lund University, Box 157, SE-221 00 Lund, Sweden

**Keywords:** anxiety, causality, depression, depressive symptoms, diet, mental health, prevention, public health, public health professionals

## Abstract

Background: Mental illness is one of the fastest rising threats to public health, of which depression and anxiety disorders are increasing the most. Research shows that diet is associated with depressive symptoms or depression (depression). Aim: This study aimed to investigate the diets impact on depression, by reviewing the scientific evidence for prevention and treatment interventions. Method: A systematic review was conducted, and narrative synthesis analysis was performed. Result: Twenty scientific articles were included in this review. The result showed that high adherence to dietary recommendations; avoiding processed foods; intake of anti-inflammatory diet; magnesium and folic acid; various fatty acids; and fish consumption had a depression. Public health professionals that work to support and motivate healthy eating habits may help prevent and treat depression based on the evidence presented in the results of this study. Further research is needed to strengthen a causal relationship and define evidence-based strategies to implement in prevention and treatment by public healthcare.

## 1. Introduction

Depression and other mental health conditions are on the rise globally and remain a threat to public health of which depression and anxiety disorders are increasing the most [1]. The prevalence of depression is in excess of 300 million people. Depression is the leading cause of nonfatal disease burden [2]. Suicide is the second leading cause of death in young adults, and nearly 800,000 commit suicide every year worldwide [1]. According to The Organization for Economic Co-operation and Development (OECD), the cost of poor mental health in the European Union (EU) was estimated to be more than 4% of gross domestic product (GDP), which corresponds to over € 600 billion. Mental health is crucial for learning, productivity and community participation, and scientific evidence supports that measures are required in the form of investment in preventive care [2]. Furthermore, research shows that lifestyle habits are associated with mental health [3]. In 2015, the United Nations General Assembly decided on new global development goals [4]. The global development goals were named Agenda 2030 and the aim of the declaration is to promote a sustainable future and eradicate global poverty. One of the sub-goals is regarding preventive measures and treatments to reduce deaths due to noncommunicable diseases such as diabetes and cardiovascular disease, as well as the promotion of mental health and well-being. Depression and other mental conditions may affect all people, and the line between mental health and illness is not easy to define [1]. Many employments in today’s society are less physically strenuous due to technological developments, while the psychological strain has increased. Mental illness for the individual can lead to problems with coping with their daily lives such as family and work [1]. Feelings of guilt and shame may be an obstacle to seek help for mental illness. Mental illness is no longer as stigmatized as it has been, and more people are seeking help from the healthcare system. Depression is one of the major public health challenges and in 2015, the World Health Organization (WHO) established a plan for how this should be addressed in “The European Mental Health Action Plan” [1]. This action plan covers mental health and mental disorders across the life-course and is fully aligned with the values and priorities of the new European policy framework for health and well-being, Health 2020.

Through evidence of a positive impact on depressive symptoms or depression (from here on depression) through dietary changes, resources may be motivated to prevent and promote healthy lifestyle habits. With reference to the costs that mental illness entails, it should be of interest for public health prevention that can lead to a reduced cost burden for the society. In a study by Berk et al. [3], existing research was compiled of lifestyle habits in managing depression in a narrative review. Their analysis resulted in three clinical recommendations. The first was that lifestyle factors such as unhealthy eating, inactive lifestyle, smoking, and substance use contributed to an increased risk of depression, and several of the factors simultaneously strengthen the relationship with lifestyle. The second clinical guideline was based on the evidence that physical exercise is effective as a treatment for depression. The third guideline was based only on a few studies and still demonstrating the effect of providing support for smoking cessation and dietary recommendations as a depression treatment. In the view of the studies showing a relationship, Berk et al. [3] highlighted the factors that may be important to avoid the risk of depression but requires more research. At the present, more studies have been published in the field, and to contribute to the knowledge base regarding lifestyle habits, the current literature study includes studies published after Berk et al. [3], and specifically regarding diet and nutrition. Mental illness is a growing threat to public health [1] and depression and anxiety disorders are the conditions that increase the most. Therefore, in the present study, mental illness was limited to encompassing these conditions. Through identified evidence, the public health professionals may justify spending resources on health promotion of eating habits as a supplement to medical treatment. The aim of this systematic review was to investigate the diets impact on depression by mapping available evidence for preventative and treatment interventions.

## 2. Materials and Methods

This literature study was conducted as a systematic review and narrative synthesis analysis was performed [5,6,7,8,9]. The literature searches were conducted systematically, and based on a narrative synthesis and meta-analysis published by Berk et al. [3] that included studies published through 2012-08-01. Data collection was conducted in the databases CINAHL, PubMed, and PsycINFO. The PICO model was used in this study [10]; P = depressive symptoms or depression; I = dietary interventions; C = exposure/intervention or no exposure/intervention; and O = preventing and treating depressive symptoms or depression.

Inclusion criteria were studies published 2012-08-01–2019-08-31, peer-reviewed empirical quantitative studies, that illustrated the aim of current study, and was conducted in Europe, Oceania, and the United States of America (USA). All included studies were approved according to research ethical standards. Eligible study design for inclusion were randomized controlled trials (RCTs) and observational studies. The countries should have comparable national dietary recommendations and discuss potential confounding factors. Studies regarding pregnant women and studies in languages other than English were excluded. The search was performed using Boolean terms such as AND, OR, and with MeSH terms and major topics (MM). The following search terms were used: Mental health; depression; anxiety; eating; diet; eating behaviors; causality; and risk factors. An example of search strategy in PubMed is presented in Table 1 according to the PRISMA checklist.

The authors conducted relevance and quality assessment individually and independently. The relevance assessment was carried out in two steps. In step one, a rough screening was performed based on titles and abstracts and in case of doubt, the articles were passed on to the next review, step two. In step two, the articles were reviewed in full text for relevance [7,11,12,13,14]. Studies considered relevant were included in the systematic literature compilation and proceeded to quality assessment. Included studies were quality assessed according to relevant review templates from the Joanna Briggs Institute [15]. To assess the quality of the reviewed studies, a quality evaluation was conducted, and the studies were assessed in relation to the categories: Strong; moderately strong; and limited (article overview, Table 2). Criteria used in the assessment were: Ethical consideration; population size; selection; drop-out; measurement methods and consideration for confounders; internal and external validity; reliability; and generalizability. Studies that did not consider the mentioned points to a satisfactory extent were assessed with low-quality corresponding to the category insufficiently and were not included in this study. A flowchart was used to provide a clear overview of the selection process of included articles, see the flowchart PRISMA [16] (Figure 1).

The searches resulted in 21 articles, in addition, one article was found by free text search in PubMed, which resulted in a total of 22 articles included in the current study (Table 1). A narrative synthesis analysis was conducted. In the analysis, the results were assessed in detail and in the synthesis, the results of the different studies were combined to create a new perspective and approach. This was initially done individually by the authors, followed by a discussion to reach consensus. The final step in the evaluation was an assessment of how strong the overall scientific basis was. The Grading of Recommendation Assessment, Development and Evaluation (GRADE) [39,40,41] system was used to assess the quality of the articles as “strong” evidence value (*n* = 11), “moderately strong” evidence value (*n* = 6), or with “limited” evidence value (*n* = 5) in preliminary evidence strength (Table 1). Studies included in this literature study have been approved on the base of each country’s current rules regarding research and ethical approval. All participants in the included articles had given informed consent to participate.

## 3. Results

The result included 22 articles, research that studied the relationship between diet and depression. The studies were conducted in Australia (*n* = 5), Canada (*n* = 2), Finland (*n* = 2), France (*n* = 6), Italy (*n* = 1), the Netherlands (*n* = 1), the United Kingdom (UK) (*n* = 1), and USA (*n* = 4). All included countries had comparable national dietary recommendations. The total number of participants in the 22 studies were 455,781 people with a variance between 56–90, 380 participants (Table 1). Studies with significant results had a *p*-value *>* 0.05. All included studies had taken into account potential confounding factors, such as age, gender, marital status, education, income, occupation, physical activity, smoking, alcohol, and body mass index (BMI).

### 3.1. Subsection

The result was divided into five categories:Adherence to dietary recommendations and risk of depression;Pro-inflammatory diet and depression;Dietary intake of folic acid, magnesium, and fatty acids linked to depression;Dietary choices and risk of depression;Causal link between diet and depression.

#### 3.1.1. Adherence to Dietary Recommendations and Risk of Depression

High adherence to dietary recommendations was in several studies showing a significant protective effect against depression and depressive symptoms, and studied in different populations based on origin, age, and gender [19,21,24,25,26,30,31,36]. High adherence to dietary advice was associated with a significantly reduced risk of developing depressive symptoms in first-time mothers in Australia [30]. High adherence to dietary recommendations was associated with a lower risk of depressive symptoms in adults in France. The adherence was measured regarding four scores namely; modified French Programme National Nutrition Sante’-Guideline Score (mPNNS-GS), Probability of Adequate Nutrient Intake Dietary Score (PANDiet), Diet Quality Index-International (DQI-I), and Alternative Healthy Eating Index-2010 (AHEI-2010) [19]. Risk reduction was in this study estimated to be 21%, 20%, and 12%, respectively, and in the AHEI-2010 group no significance was detected. High adherence to French recommendations on diet and physical activity showed a highly significant association with reduced risk of depression [24]. Adherence to different dietary recommendations has been shown to be effective in achieving improved mental health in middle-aged and older people with current or past depression diagnosis [25]. The relationship was significant among men, where poorer adherence to a Mediterranean diet and AHEI were associated with a significantly higher risk of depression. Adherence to the Dietary Approaches to Stop Hypertension (DASH) diet showed no significant association with depression [25]. AHEI has shown a significant association with protective effects against depression in women [21]. High intake of fruits, vegetables, fiber, and low intake of trans fats were in the same study associated with a lower risk of recurrence in depression. An analysis of dietary points based on AHEI revealed a dose-response relationship that was significant for women. The women who improved their AHEI scores over 10 years had a 65% lower risk of recurrent depressive symptoms compared to the women who continued to have low AHEI scores. In the study, no significant results were found in men [21]. In a study from Finland, a healthy diet with a higher proportion of vegetables, fruit, chicken, fish, whole grain products, legumes, berries, and low-fat cheese was compared to a typically Western diet including processed foods such as sausages; French fries; fast foods; sweets such as ice cream, cakes, candy, and chocolate; soda; processed meat; baked potatoes; high-fat cheese; and eggs [36]. In the studied population of middle-aged Finnish men, a significant protective effect was found with a higher degree of adherence to a healthy dietary pattern. The protective effect was measured at a 25% reduced risk of suffering from depressive symptoms. A significant risk increased for depressive symptoms with a diet consisting of a Western diet was measured at a 41% increase in the same study. Recommendations of a healthy diet and intake of fruits and vegetables have shown significant protective effects for mental illness among immigrants in Canada and a higher intake of fruit and vegetables was associated with 19%–23% of improved mental health [26].

#### 3.1.2. Pro-Inflammatory Diet and Depression

Several studies showed an association between dietary intake with inflammatory potential and risk of depression in different populations [17,18,20,22,33]. Products associated with less impact on systemic inflammation have been found to be vegetables, whole grains, olive oil, and fish. Products such as sweets; refined flour; high-fat products; red and processed meat were associated with a greater impact on systemic inflammation [17]. The results showed that a pro-inflammatory diet was associated with a significantly increased risk of depression in the subgroup of women; middle-aged adults; and people with overweight and obesity. Thus, the relationship was strongest in people with overweight and obesity [17]. An increased risk of depression was associated with a high proportion of processed foods in the diet, and for each 10% increase of the proportion of processed foods [18]. High intake of pro-inflammatory food was associated with significantly increased risk of depressive symptoms [17,18,20,22]. In subgroups of men, smokers and physically inactive, a diet consisting of a higher proportion of pro-inflammatory foods, significantly increased the risk of depressive symptoms [20]. Associations between food with inflammatory effect and increased risk of depression were calculated with significance in a cross-sectional study performed in USA [22]. A high intake of inflammatory diet was significantly associated with the occurrence of frequent anxiety in the same study. In another study from USA, the results indicated a significant association between inflammatory diet and risk of depression in women [33].

#### 3.1.3. Dietary Intake of Micronutrients Linked to Depression

Micronutrients in the diet have been associated with an increased risk of mental illness [17,19,23,28,29,32,38]. Magnesium intake through diet was significantly associated with the risk of developing depression in middle-aged men [38]. Calculations were made between three statistical models regarding the content of magnesium in the diet, and the lowest intake of magnesium was associated with a significantly increased risk of depression. When all three models in the study were compared, it was found that those with the highest magnesium intake had a protective effect against depression [38]. Relationships between intake of B12, folic acid, and magnesium emerged as side effects in the result where the main purpose was to investigate adherence to healthy dietary advice [19]. Those with the highest adherence to healthy dietary advice in the same study had thus less risk of depressive symptoms and a significantly higher intake of magnesium, folic acid, and B12 in the diet. In another study, high intake of processed food increased the risk of depression, and those with high intake of processed food had significantly lower intake of B12, magnesium, and folic acid in their diet, compared to the group that had the lowest intake of processed foods [18]. Significant associations in both genders have been calculated regarding vitamin B intake and the risk of depression [29]. In women, those with the highest intake of B6 had a reduced risk and among men, the ones with the highest intake of B12 had a reduced risk. Low levels of B6 and B12, respectively were associated with an increased risk of depression in the same study. Intake of fatty acids in the diet was investigated as mediators on risk of inflammation and associations with depression in older people [32]. Inflammation markers in the study were measured with C-reactive protein (CRP) and interleukin-6 (IL-6). The result showed that omega 3 and polyunsaturated fatty acids had protective effects for depression in men, and CRP was the marker that was significantly affected. Furthermore, the total intake of fat, saturated fatty acids, and monounsaturated fatty acids had a significantly increased impact on both CRP and IL-6 in women. Dietary intake of flavonoid subclasses had in one study, a significant protective effect against risk of depression among women. Futhermore, the highest intakes of flavonols, flavones, and flavanones were significantly associated with a 7%–10% lower risk of depression compared with the lowest intakes in a study from USA [23]. In a study by Godos et al. [28], dietary intakes of total polyphenols, their classes, subclasses, and compounds were assessed in relation to depressive symtoms. In their result, no significant association with depressive symptoms was found with the total polyphenol intake. In subclasses, this study assessed significance, indicating that higher flavonoids intake may be inversely associated with depressive symptoms.

#### 3.1.4. Dietary Choice and Risk of Depression

Different diets have been associated with increased or decreased risk of depression [17,18,27,34,37]. Relationships have been demonstrated with the risk of developing depression when excluding specific food from the diet [34]. The prevalence of depressive symptoms was highest for vegans (28.4%) and lowest among omnivores (16.2%). The study showed that regardless of diet, the risk of depression increased significantly with the number of foods excluded from the diet [34]. A dose-response relationship emerged in young women regarding fish consumption and the risk of depression, and the trend was a 6% reduction per serving of fish [37]. Women who ate fish twice a week or more had a 25% lower risk of depression during the follow-up period compared to women who ate fish less than twice a week. Women who developed depression during the follow-up period were 15% less likely to eat fish twice a week compared to women who did not develop depression. Fish intake was not significantly associated with depression in men [37]. Fish consumption was included as a variable in one study where pro-inflammatory diet and risk of depression symptoms were studied in men and women [17]. With a high intake of pro-inflammatory food, a significantly increased risk of depression symptoms was seen. At the same time, those with the highest risk had a significantly lower intake of meat, fish, and eggs compared to those who ate a more unprocessed diet, and this relationship applied to both men and women [17]. The dietary glycemic index (GI) was associated with a significantly increased risk of depression in postmenopausal women in the USA [27]. In the same study, a higher intake of added sugar was associated with a significantly increased risk of depression, and high consumption of fruits, vegetables, and fibre was associated with a significantly reduced risk of depression.

#### 3.1.5. Causal Link between Diet and Depression

The result included two randomized controlled trial (RCT) studies [31,35] that we are able to demonstrate a causal relationship between diet and mental illness. Individual counseling with motivational interviewing (MI) for increased adherence to a healthy diet yielded a significantly better treatment outcome compared to conventional treatment (social support group) [31]. The duration of counseling conversations according to MI and social conversation support were the same in both groups. Those who received support in dietary change based on MI showed significantly reduced depression symptoms, and the strength of the result was calculated to the number needed to treat (NNT) 4 [31]. Adherence to the Mediterranean diet led to significantly improved mental health in adults with depression [35]. The intervention group that received support and group training in eating according to the Mediterranean diet had a significantly higher intake of vegetables, fruits, whole grains, and nuts and a significantly lower intake of sweets than the control group who received group meetings with social activity. In the dietary intervention group, a 1.68-fold reduction in depression symptoms was calculated, which remained during the six-month follow-up period [35].

## 4. Discussion

The results of this study showed that diet can be important for the emergence and treatment of mental illness. High adherence to dietary recommendations, anti-inflammatory diet, fish consumption, exclusion of processed foods, and adequate intake of folic acid, magnesium different fatty acids, were associated with a reduced risk of mental illness.

Adherence to recommended dietary advice generates adequate intake of nutrients and can reduce risk and reduce symptoms of mental illness. The results of this study showed a dose-response relationship with following current dietary recommendations and reduced risk of mental illness in different ages and populations. This result is consistent with recently published systematic literature studies showing a consistent result. A balanced diet with high intake of vegetables, fruits, and fish was associated with reduced risk of depression, whilst a diet with added sugar, soda, and junk food was associated with increased risk of depression [42,43,44]. The results of the present study and other recently published results strengthen Berk et al. [3] results and thus their presented guidelines. Dietary advice recommended by authorities and WHO should be followed as good adherence has been shown to be associated with a protective effect against mental illness.

Food may have an influence on the level of inflammation in the body and may be linked to increased risk of depression and food with anti-inflammatory effect can reduce symptoms and protect against mental illness. This study showed that the anti-inflammatory diet reduced the risk of depression and depression symptoms, and conversely, a high intake of food with pro-inflammatory effect increases the risk. These findings are supported by studies from other countries where the prevalence of depression and anxiety was associated with a high intake of pro-inflammatory diet [45]. In another study, the inflammatory potential of the diet was investigated in teenage girls, and the results showed significantly increased stress levels in teenage girls when the diet consisted of a high proportion of pro-inflammatory food [46]. Lassale et al. [43] meta-analysis demonstrated that avoiding a pro-inflammatory diet was associated with a lower risk of depression, supporting the results of the present study. The available research on the relationship between inflammation, intestinal flora, and mental illness such as depression and anxiety has been studied in systematic literature [47]. The result showed that probiotics (intake of good bacteria) and prebiotics (food for the good bacteria) lead to improved intestinal health, which can reduce inflammation and thus result in fewer symptoms of mental illness [47]. Foods with a pro-inflammatory effect should be avoided for preventive purposes before and in case of mental illness.

The diet’s various constituents of micronutrients can be of importance for depression. Fatty acids in the diet can affect the level of inflammation in the body and have been linked to increased risk of depressive symptoms. The intake of micronutrients showed in this study a protective effect against depressive symptoms. Higher intake of magnesium, folic acid, B6, and B12 through the diet had protective effects against mental symptoms in different populations. The diet’s content of fatty acids was important for mental illnesses due to increased inflammation, different fatty acids increased, or decreased the risk of depression. These findings in our study are supported by studies in older people where significant associations were found between low levels of folate and B12 in serum and increased risk of depression in both sexes [48], and a high dietary intake of B6 and B12 showed significant protective effect for the development of depression [49]. Furthermore, a low intake of magnesium in the diet has also been associated with increased risk of depression in populations of younger and older persons in other studies [50,51]. Previous studies and a meta-analysis [52,53] found significant correlations with folic acid levels and depression, where low blood levels were associated with increased risk and a high diet-based intake of folic acid with a protective effect against depression. A diet with an adequate intake of micronutrients may protect against depression and should be promoted in all ages.

Exclusion of foods from the diet may be associated with an increased risk of mental illness, as well as the intake of refined foods with high GI such as sweets, soft drinks, and whole foods. The results in this study showed that healthy diet choices such as increased fish consumption and the Mediterranean diet may have a protective effect, while a high intake of sweets and products with high GI were associated with increased risk of depressive symptoms and depression. Furthermore, regardless of diet, the risk of depression increased with the number of foods excluded from the diet. Several studies and meta-analyses have shown that people whose diet consisted of a Western diet had a significantly increased risk of depression, while those who instead ate a lot of vegetables and whole grains had a significantly reduced risk [54,55]. In another literature review, the consumption of sweetened beverages such as soft drinks and the association with the risk of depression was investigated. The result showed a significant increase in risk of depression in people with a high intake of soft drinks [56]. Fish consumption has been associated with a reduced risk of depression in several published articles [57,58,59]. In the most recently published of these reviews, Yang, Kim and Je [57] found a significant dose-response relationship, which strengthens the results of the present study. A varied diet rich in nonrefined foods, as well as fish can protect against depression.

Dietary interventions may be an effective treatment for depression. Support for dietary changes can consist of MI and group interventions with education and advice on healthy eating, which may provide effective treatment and thus, lead to a reduction of depressive symptoms. The results of this study showed a significant positive effect on the mental health of those who received dietary intervention in the form of education regarding nutrition and cooking at a group level and MI-based guidance on an individual level. Furthermore, the result included a few RCT studies with dietary interventions compared with social conversation support or group activity such as treatment for depression. These results showed a significant difference and a causal relationship between the importance of diets for mental health. Research in The MooDFOOD project is currently forthcoming, and in a close future, evidence of the importance of diet may be clarified [60]. In one of the RCT studies conducted and included in the results of the present study, NNT was 4, and if this result will remain in future studies with dietary interventions, this could constitute a powerful treatment for depressive symptoms and depression. In order to provide a perspective for NNT 4, it may be compared to statins that are now used for primary prevention, and NNT has been calculated at 104 for preventing myocardial infarction and NNT 154 for preventing stroke [61]. In an RCT study by Estruch et al. [62], the preventive effect of these drugs against heart disease and stroke, compared to treatment with the Mediterranean diet and based on this dietary intervention for primary prevention of myocardial infarction and stroke, the result was NNT 61 [62]. These results do not specifically highlight mental illness but indicate that diet may have a major impact on health. Guidance and counseling at individual and group level may be effective in preventing and treating mental illness. Inflammation levels in the body have been linked to several disease states, and in our results also to mental illness [17,18,20,22,33]. The global prevalence of the metabolic syndrome is high (over 1 trillion people), and metabolic syndrome leads to prediabetes, high blood pressure, and increased levels of inflammation [63]. This connection is logical since mental health is not separated from physical health since everything is interconnected. A large part of the mental illness of our time may be prevented when the cause is based on unhealthy lifestyle habits. Health care professionals need to work preventively in a larger extent and treat causes rather than the symptoms [2]. The RCT studies and the high-quality cohort studies with strong evidence included in the results of the present study strengthen the results. Based on the weighting of knowledge that emerges from the results of the present literature study, the development of Berk et al. [3] dietary guidelines may be defined. Furthermore, this knowledge can be used in public health professionals’ health-promoting work, preferably in combination with MI at an individual and group level. The basis of RCT studies with dietary counseling interventions to support and motivate people to change their dietary habits is limited, and those that exist have shown a positive effect. Further research is needed to strengthen evidence of effective strategies aimed at supporting people’s change in dietary habits for the prevention and treatment of depressive symptoms or depression.

The PICO method was used to compile search strategies based on the aim of this study for increased accuracy in the databases and to establish internal validity. The words “causality” and “risk factors” were used in free text in CINAHL and PsycINFO, since searching through Major topics led to a restriction. To avoid major cultural differences and to achieve a more generalizable and transferable result, the included studies were conducted in Western countries. The publication years included the most recent research after the publication of Berk et al. [3], July 2012 until 2019 September, which strengthens the timeliness of the result. To strengthen the ethical aspect, only studies with ethical approval were included. Pregnant women were excluded, as this is a group of patients that primarily meet midwives, and that pregnant women have hormone effects, which could affect the result.

The authors conducted assessment for relevance and review of the articles for quality individually, using validated templates (Joanna Briggs). The articles were discussed for consensus to strengthen reliability. Confounding factors may be a problem in observational studies and may be controlled in different ways. All studies included in the result were checked for a large number of confounding factors. As an example, most included studies have stratified the material into subgroups and used regression models. The evaluation of the study quality according to GRADE includes confounding factors and studies that have not taken this into account have not been included in this study which is a strength. The synthesis of the results was carried out first individually and then in a joint discussion which opened up an independent interpretation and division into the different categories. The number of articles in this literature study may be seen as a strength. The results from the various articles are similar, which increases transferability.

One weakness of this study is that few RCT studies were included in the result. More RCT studies may have strengthened the result and causality. Regarding results with risks, it is important to include observational studies, since RCT studies are usually not able to answer more long-term questions. Observational studies can provide important information and contribute to the overall evidence evaluation. Included studies in this systematic review are well-conducted observational studies where large cohorts have been followed over time. Another aspect of the transferability of results is that studies conducted in other parts of the world with different cultures show a result in line with the present study. Studies conducted in China and Iran have shown that a healthy diet has a positive impact on mental health in the same way as the results of the present study. This supports the fact that regardless of culture or population, these are health-promoting factors in the diet that strengthen the results of the present study.

## 5. Conclusions

The diet may have a significant effect on preventing and treating depression for the individual. A diet that protects and promotes depression should consist of vegetables, fruits, fibre, fish, whole grains, legumes and less added sugar, and processed foods. In the public health nurse’s preventative and health-promoting work, support and assistance with changing people’s dietary habits may be effective in promoting depression.

Based on the results of the present systematic review, the following may be recommended:-Advice should be given to people regarding health-promoting diets such as increasing the intake of vegetables, fruits, fish, nuts, legumes, olive oil and excluding or severely restricting the intake of processed foods such as sausages, juices, soft drinks, and sweets to promote and prevent depression (strong evidence value).-The diet’s content of micronutrients such as magnesium, folic acid, and various B vitamins is of importance for depression (moderately strong evidence value).-The public health professionals may advantageously use MI as an approach and strategy in the guiding work to motivate and support dietary changes in depression (limited evidence value). Resources should be used to help people maintain a healthy diet for preventive purposes for depression (strong evidence value).

## Figures and Tables

**Figure 1 ijerph-17-01616-f001:**
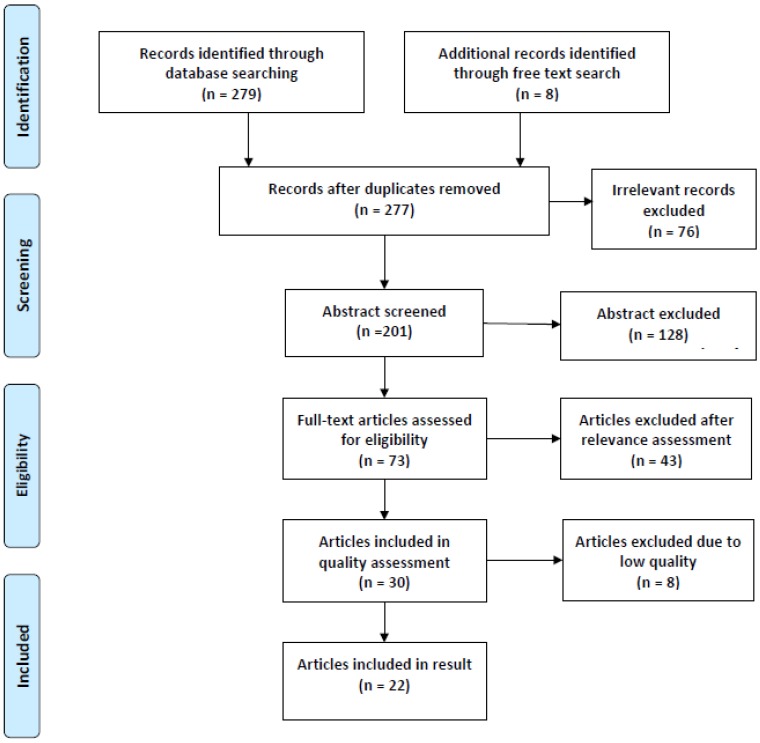
Flow chart of included articles in the study [16].

**Table 1 ijerph-17-01616-t001:** Search and search terms for PubMed.

Search Number (#)	Search Terms	Hits	Reviewed Abstracts	Selection 1 Relevance Assessment	Selection 2 Quality Assessment	Selection 3 Included Studies
#1	Eating [MeSH] OR Diet [MeSH]	324,299	-	-	-	-
#2	Causality [MeSH]	796,445	-	-	-	-
#3	Depression [MeSH] OR Anxiety [MeSH] OR Mental health [MeSH]	288,337	-	-	-	-
#4	#1 AND #2 AND #3	175	127	47	20	14

Limitations: 2012-08-01–2019-08-31, English. The search was performed 2019-08-31. MeSH: Medical Subject Headings.

**Table 2 ijerph-17-01616-t002:** Overview of included articles in the study.

Author/ Year/ Country	Study Design/ Number of Participants	Population/ Selection	Aim	Results	Conclusion	Study Quality According to GRADE
Adjibade et al. (2019a) [17] France	Prospective cohort *n* = 26,730	Participants from Nutrinet-Santé Adults 18–86 years	Relationship between inflammatory diet and depressive symptoms.	The result indicates that a pro-inflammatory diet may be associated with an increased risk of depression symptoms, especially in obese and obese participants (Hazard Ratio (HR) 1.29).	Maintaining normal weight and avoiding pro-inflammatory foods can be interesting targets for prevention strategies to prevent depression and depressive symptoms.	Strong
Adjibade et al. (2019b) [18] France	Prospective cohort *n* = 26,730	Participants from Nutrinet-Santé Adults 18–86 years	Relationship between processed food and risk of depression.	An increased risk of depressive symptoms was observed in a high proportion of processed foods in the diet (HR 1.21).	Avoiding processed foods can be important in preventing mental illness.	Strong
Adjibade et al. (2018) [19] France	Prospective cohort *n* = 26,225	Participants from Nutrinet-Santé Adults 18–86 years	Examine the prospective relationship between adherence to dietary advice and the incidence of depression.	High adherence to dietary advice was associated with lower risk of depressive symptoms.	Adherence to dietary advice can have a preventive effect on depression.	Strong
Adjibade et al. (2017) [20] France	Prospective cohort study mean follow-up 12.6 years *n* = 3523 35–60 years	Participants from SU.VI. MAX cohort that initially did not have depressive symptoms Adults	Investigate prospective relationship between diet’s potential inflammatory effects and risk of developing depressive symptoms.	Significance was found in the subgroups of men, smokers, and physically inactive that a diet with a higher proportion of pro-inflammatory foods could increase the risk of depressive symptoms (Odds Ratio (OR) 2.32 / 2.21 / 2.07, respectively).	Promoting a healthy diet consisting of anti-inflammatory foods can help prevent depressive symptoms.	Strong
Akbaraly et al. (2013) [21] United Kingdom	Prospective cohort *n* = 4215	Data from Whitehall II Adults 35–55 years	Investigating whether adherence to a healthy diet was associated with depressive symptoms.	A healthy dietary index was associated with protective effects against depression in women.	For women, a healthy diet can prevent depression.	Moderately strong
Bergmans and Malecki (2017) [22] USA	Cross-sectional study *n* = 11,592	Data from NHANES Adults aged 20 years and older	Investigate whether the diet’s inflammatory index is associated with depression and other mental illness.	A high intake of inflammatory foods was associated with significantly increased risk of depression (OR 2.26), and for frequent anxiety (OR 1.81).	A pro-inflammatory diet has the potential to increase inflammation, which in turn can increase the risk of depression.	Limited
Chang et al. 2016 [23] USA	Prospective cohorts with 10 y of follow-up Associations *n* = 82,643	Data from Nurse health study Women 36–80 years	Greater intakes of dietary flavonoids were significantly associated with a modest reduction in depression risk, particularly among the older women.	Higher intakes of all flavonoid subclasses except for flavan-3-ols were associated with significantly lower depression risk; the strongest associations (HR for both: 0.83) was found for flavones and pro-anthocyanin showed.	Higher flavonoid intakes may be associated with lower depression risk, particularly among older women.	Strong
Collin et al. (2016) [24] France	Prospective cohort *n* = 3328	Data from SU.VI.MAX study Adults 35–60 years	Investigate whether there is a relationship between adherence to dietary recommendations and depression in middle-aged women and men.	Adherence to French dietary advice was associated with lower incidence of chronic and recurrent depressive symptoms (*p* > 0.001).	Adherence to dietary recommendations may be relevant to avoiding depression symptoms.	Strong
Elstgeest et al. (2019) [25] The Netherlands	Longitudinal cohort study *n* = 1439	Data from the LASA Nutrition and Food-related behaviour Adults 55–85 years	Investigate the relationship between depression and diet quality.	Current and past depressive symptoms were associated with poorer dietary quality based on dietary index, especially in men.	Men with depression symptoms can feel better by eating high quality diets.	Limited
Emerson and Carbert 2019 [26] Canada	Prospective cohort cross-section measurement *n* = 37,071	Data from the Canadian Community Health Survey From 12 years.	Investigate the relationship between nutrition and depression of immigrants in Canada.	Fruits and vegetables showed significant protective effects against depressive symptoms such as mood and anxiety disorders.	A healthy diet should be considered to prevent mental illness for immigrants in Canada.	Limited
Gangwisch et al. (2015) [27] USA	Prospective cohort *n* = 87,618	Data from Women’s Health Initiative Observational study Women 50–79 years.	Investigate the association between high GI and depression in women post menopause.	Dietary glycemic index (GI) was associated with a significantly increased risk of depression in postmenopausal women (OR 1.22). A higher intake of added sugar was associated with a significantly increased risk of depression (OR 1.23) and high consumption of fruits (OR 0.88) vegetables (OR 0.88), and fibre (OR 0.86) was associated with a significantly reduced risk of depression.	High GI diets may be a risk factor for depression in postmenopausal women.	Strong
Godos et al. 2018 [28] Italy	Observational study *n* = 1572	Data from the MEAL study Adults 18–92 years.	To assess the relation between estimated dietary intakes of total polyphenols, their classes, subclasses and compounds, and depressive symptoms.	Total polyphenol intake was not associated with depressive symptoms. However, for certain polyphenol classes intake was significant inverse associated with depressive symptoms, comparing intake from the highest with the lowest quartile. Phenolic acid (OR 0.64), flavanones (OR 0.54), and anthocyanins (OR 0.61).	Greater dietary intake of flavonoids may be inversely associated with depressive symptoms.	Moderately strong
Gougeon et al. 2016 [29] Canada	Longitudinal observational study *n* = 1793	Data from a randomized medical database in Quebec Adults 67–84 years.	Relationship between intake of B vitamins and depression.	In women, those with the highest intake of B6 had a reduced risk (OR 0.57) and among men, the ones with the highest intake of B12 had a reduced risk (OR 0.42).	Intake of vitamin B can affect the risk of depression.	Strong
Huddy et al. (2016) [30] Australia	Cross-sectional study *n* = 437	Data from the Melbourne Infant Feeding, Activity, and Nutrition Trial Women 19–45 years	Examine the relationship between intake of fruits, vegetables, fish, and depression in first time mothers.	Healthy diet was associated with less depressive symptoms.	Adherence to a healthy diet can reduce the risk of depressive symptoms in first time mothers.	Limited
Jacka et al. (2017) [31] Australia	RCT *n* = 56	Single blind RCT Adults from 18 years	The aim was to investigate the effectiveness of diet improvement programs to treat depression.	Both groups had a significantly better condition after three months. The group receiving treatment with dietary intervention had more than 30% improved condition, compared to those receiving social support 8%.	Treatment with dietary interventions in the form of individual counselling according to MI can help people with depression to improve mental health.	Strong
Lai et al. (2016) [32] Australia	Prospective cohort *n* = 2035	Data from the Hunter Community study. Adults 55–85 years	Inflammatory markers influence on antioxidant and fatty acid intake is there an association with depression.	Relationships were observed between the intake of various fatty acids, antioxidants, the influence of inflammatory markers, and the risk of depression.	There may be a link between fatty acid intake and depression, partly mediated by inflammatory markers.	Moderately strong
Lucas et al. (2014) [33] USA	Prospective cohort *n* = 43,685	Data from the Nurses’ health study Women 55–77 years	The aim was to investigate the relationship between inflammatory diet and depression.	An inflammatory diet was significantly associated with an increased risk of depression (Reduced Rank Regression (RRR) 1.41 (*p* > 0.001).	Chronic inflammation may be an underlying link between diet and depression.	Moderately strong
Matta et al. (2018) [34] France	Prospective cohort Cross-sectional *n* = 90,380	Data from Constances Cohort Adults 18–69 years	The study examined the cross-sectional relationship between depressive symptoms and vegetarian diet.	Regardless of diet, the risk of depression gradually increased with the number of excluded foods.	Depressive symptoms can be associated with food exclusion.	Limited
Parletta et al. (2019) [35] Australia	RCT *n* = 85	Single blind RCT Adults 18–65 years	The purpose was to investigate whether the Mediterranean diet supplemented with fish oil could improve the mental health of people with depression.	The group that received the intervention with dietary advice and education according to the Mediterranean diet had significantly improved mental health after three months and the positive result remained after six months follow-up.	Healthy dietary changes are possible and have been shown to improve the mental health of people with depression.	Strong
Ruusunen et al. (2014) [36] Finland	Prospective cohort *n* = 1003	Data from Kuopio Ischemic heart disease Risk Factor Men 46–65 years.	Assess diet’s relationship with depression.	Three different dietary patterns were identified. The group with the highest intake of a healthy diet had a protective effect against depression.	Adherence to a healthy diet can reduce the risk of depression.	Moderately strong
Smith et al. (2014) [37] Australia	Longitudinal study *n* = 1386	Data from The Childhood Determinants of Adult Health Adults 26–36 years	Relationship between fish consumption and risk of depression.	Fish consumption significantly protected against depression in women, in men no relationship was seen.	Regular fish consumption can reduce the risk of depression in women.	Strong
Yary et al. (2015) [38] Finland	Prospective cohort study 20 years of follow-up *n* = 2320	Data from The KIDH study Men 42–61 years	Relationship between magnesium intake in the diet and risk of depression.	Magnesium intake had a protective effect against depression (HR 0.53).	Magnesium intake may have an impact on the risk of developing depression in men.	Moderately strong

GRADE: The Grading of Recommendation Assessment; GI: Glycaemic Index; KIDH: Koupio Ischemic Heart Disease risk factor study; LASA: Longitudinal Ageing Study Amsterdam; NAHNES: National Health and Nutrition Examination Survay; RCT: Randomized Controlled Trial; SU.VI.MAX: Supplémentation en Vitamines et Minéraux Antioxidants; USA: United States of the America.

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
