# Peer review of "Evidence of the Importance of Dietary Habits Regarding Depressive Symptoms and Depression"

_ijerph, 2020, doi:10.3390/ijerph17051616_

Round 1

Reviewer 1 Report

In a very synthetic way, i present my appreciation for the work that has been endorsed to me:

In abstract: This work is a systematic review, not a narrative review.

In line 61: the cited article is not a meta-analysis.

In line 83 to 84: I didn't understand the use: “C=the importance of diet for mental health/mental illness”, The C represents the comparison group, when using PICO, assuming we are only researching clinical trials.

In line 84 to 85: The O is only “preventing and treating mental illness”.

In line 89 to 90: Why did they use this time limit, with an interval in the middle: “Exclusion criteria were studies published before 2012-08-01 and after 2019-08-31”. Usually the time limit of the research is placed as an inclusion criterion.

In line 92: The title of the work refers to mental illness, in general. Why did they particularize and use the terms “depression” and “anxiety” in their research?

In line 93: Because the terms “prevention” and “treatment” were not used, as defined for the "Outcomes"?

In Figure 1: In flow chart it is necessary to explain the "why" of the articles that were eliminated by reading the full text.

In Table 1: From what can be seen, most studies focused on depression. We can call this study by: “Evidence of the importance of dietary habits regarding mental illness from public health nurses’ perspective”? I don't think so, as the results presented focus mainly on "depression".

In line 115: Where was the free text search conducted?

In line 136: Most studies address depression and not mental illness.

In line 138: In the results, it is the pro-inflammatory diet that is essentially addressed.

in the results section: It would be more interesting for the measures of association and respective levels of significance, used in the different studies, to appear in the Table 1. It would make this section "lighter" and easier to read.

In line 178: Is the pro-inflammatory diet that is essentially addressed.

In line 243: This section refers mainly to studies of dietary intervention in depression. I suggested changing this subtitle.

In line 358: Mental illness is referred to, when in reality, most studies address depression. Only reference number 25, addresses mental illness generically.

I suggested reformulating this work, directing it towards depression, eliminating the study with reference number 25.

I would also reconsider the title. Why from the perspective of public health nurses?

Author Response

Response to reviewer 1, round 1

In a very synthetic way, I present my appreciation for the work that has been endorsed to me:

  • Author’s response: Thank you very much for reviewing our manuscript and for your valuable comments for improvement. We have taken all reviewers comments into consideration and all changes in the manuscript is written in red.

In abstract: This work is a systematic review, not a narrative review.

  • Author’s response: Thank you very much. This is changed on line 18.

In line 61: the cited article is not a meta-analysis.

  • Author’s response: Thank you, this is changed on line 59.

In line 83 to 84: I didn't understand the use: “C=the importance of diet for mental health/mental illness”, The C represents the comparison group, when using PICO, assuming we are only researching clinical trials.

  • Author’s response: Thank you very much for this comment. We aren´t only researching clinical trials and the C in PICO is changed to “exposure/intervention or no exposure/intervention” on line 82. Concerning all reviewers’ comments, we have changed mental illness to depressive symptoms or depression.

In line 84 to 85: The O is only “preventing and treating mental illness”.

  • Author’s response: Thank you, this is changed in line 83.

In line 89 to 90: Why did they use this time limit, with an interval in the middle: “Exclusion criteria were studies published before 2012-08-01 and after 2019-08-31”. Usually, the time limit of the research is placed as an inclusion criterion.

  • Author’s response: Thank you and we remove this and added the inclusion period, line 84.

In line 92: The title of the work refers to mental illness, in general. Why did they particularize and use the terms “depression” and “anxiety” in their research?

  • Author’s response: Thank you for this comment. This limitation is explained in line 70-72. In collaboration with a Liberian, the keywords used in this review covers the terms depression and anxiety.

In line 93: Because the terms “prevention” and “treatment” were not used, as defined for the "Outcomes"?

  • Author’s response: Thank you for this comment. When using “prevention” and “treatment” we didn´t get any relevant hits in the searches. When using causality there were hits. This was done in collaboration with the Liberian and is further explained in the Discussion, methodological section (line 368-399), line 370-372 that unfortunately was missing in the previous version of the paper which we are very sorry for.

In Figure 1: In flow chart, it is necessary to explain the "why" of the articles that were eliminated by reading the full text.

  • Author’s response: Thank you, this is now clarified in Figure 1, page 4, line 113.

In Table 1: From what can be seen, most studies focused on depression. We can call this study by: “Evidence of the importance of dietary habits regarding mental illness from public health nurses’ perspective”? I don't think so, as the results presented focus mainly on "depression".

  • Author’s response: Thank you for this very important comment and we understand your point of view. We have changed the title to “Evidence of the importance of dietary habits regarding depressive symptoms and depression”? Line 3.

In line 115: Where was the free text search conducted?

  • Author’s response: Thank you, the free text search in conducted in PubMed as described on line 121-122.

In line 136: Most studies address depression and not mental illness.

  • Author’s response: Thank you for this very important comment and you are totally wright. We have now changed this throughout the paper.

In line 138: In the results, it is the pro-inflammatory diet that is essentially addressed. 

  • Author’s response: Thank you and this is now changed on line 145.

In the results section: It would be more interesting for the measures of association and respective levels of significance, used in the different studies, to appear in the Table 1. It would make this section "lighter" and easier to read.

  • Author’s response: Thank you for this comment, and we have now removed measures of association and levels of significance to Table 2 (previous Table 1) to make the result section “lighter”.

In line 178: Is the pro-inflammatory diet that is essentially addressed.

  • Author’s response: Thank you and you are totally right and we have now changed this to pro-inflammatory diet in line 145 and 185.

In line 243: This section refers mainly to studies of dietary intervention in depression. I suggested changing this subtitle.

  • Author’s response: This is changed on line 150, 185, 203, 234, and 255.

In line 358: Mental illness is referred to when in reality, most studies address depression. Only reference number 25, addresses mental illness generically.

  • Author’s response: Thank you for this very important comment and you are totally right. We have now changed this throughout the paper.

I suggested reformulating this work, directing it towards depression, eliminating the study with reference number 25.

  • Author’s response: Thank you for this comment and we have followed your recommendation throughout the paper. We reconsidered paper #25 since the research investigates depressive symptoms such as mood and anxiety disorders, so it´s still in the review. We also added 2 other papers suggested by another reviewer.

I would also reconsider the title. Why from the perspective of public health nurses?

  • Author’s response: Thank you and we understand your comment. This paper was from the beginning, in its previous form, targeting public health nurses and we have now removed the nurses since the paper is actually targeting all professionals in public health.

Reviewer 2 Report

Thank you for the opportunity to review this manuscript. It is a great topic of interest to providers who work with patients in mental health settings.  The current practice seems to focus a lot on the pharmacologic and counseling approach with little emphasis on diet.  

The title and the introduction: Well written and provide the context and the necessary information for the reader.  There is a minor grammatical error (see line 37 to 38).  

Methods: A little confusing for the reader, particularly as it relates to the study conducted by Berk et al.  It is not clear to the reader if the reference made in various areas of the manuscript (see lines 259, 264,275, 282-283; 323-325, among others) refers to Berk et al or to the results of the literature review conducted with the 20 studies selected.  Please clarify for the reader "the results of this study; the results of the present study" statements.  They are misleading.  Similar clarification is needed for line item 364.  

Conclusions:  This section may need a short statement indicating this is a systematic review.  Additionally, while the current review of the literature focused on diet and its role in mental health, there are other factors that predispose individuals to the development of mental health problems such as genetic factors, substance use and the like.  A concluding statement acknowledging these additional factors may assist public health nurses to discuss them with patients as well.  

Author Response

Response to reviewer 2, round 1

Thank you for the opportunity to review this manuscript. It is a great topic of interest to providers who work with patients in mental health settings.  The current practice seems to focus a lot on the pharmacologic and counseling approach with little emphasis on diet. 

  • Author’s response: Thank you very much for reviewing our manuscript and for your valuable comments for improvement. Unfortunately, the methodological discussion (line 368-399), was missing in the previous version of the paper which we are very sorry for. We have taken all reviewers comments into consideration and all changes in the manuscript are written in red.

The title and the introduction: Well written and provide the context and the necessary information for the reader.  There is a minor grammatical error (see line 37 to 38). 

  • Author’s response: Thank you very much! The grammatical error is now corrected on line 35.

Methods: A little confusing for the reader, particularly as it relates to the study conducted by Berk et al.  It is not clear to the reader if the reference made in various areas of the manuscript (see lines 259, 264,275, 282-283; 323-325, among others) refers to Berk et al or to the results of the literature review conducted with the 20 studies selected.  Please clarify for the reader "the results of this study; the results of the present study" statements.  They are misleading.  Similar clarification is needed for line item 364. 

  • Author’s response: Thank you for this comment. We have tried to be very clear throughout the paper. In the introduction, it is stated on line 67-70 that our study is contributing to knowledge after Berk 2012; and in the method section inclusion criteria’s for included studies in this study, line 84, and on line 121-122 including Table 2 (not including Berk, previous Table 1). The result section only presents included studies and refers to our result, line 67-70. It´s already clarified as you wish on line 271, 281, 287, 318-319 and 334-335. Further clarification has been made on line 307 and 408.

Conclusions:  This section may need a short statement indicating this is a systematic review.  Additionally, while the current review of the literature focused on diet and its role in mental health, there are other factors that predispose individuals to the development of mental health problems such as genetic factors, substance use and the like.  A concluding statement acknowledging these additional factors may assist public health nurses to discuss them with patients as well. 

  • Author’s response: This is now clarified on line 408.

Reviewer 3 Report

The main limitation of the study is that the search is limited to publication date, thus the study cannot provide an evidence-based information. This should be listed among the limitations of the study.

Line 92 it is necessary to mention which tool was used in order to asses study quality and provide adequate reference.

Description of inclusion/exclusion criteria should be improved (example information on the eligible study design is lacking).

Authors should provide all search terms used in the databases.

I have serious concern regarding search criteria as several studies that are fitting inclusion criteria were omitted. Authors should implement the following studies investigating the association between dietary polyphenols intake and depressive symptoms in the systematic review: PMID: 27413131, PMID: 29695122.

Table 1 Information in column “aim” should be provided as two separates columns “exposure/intervention”, “outcomes”. Moreover, authors should provide information whether results reported in the eligible studies controlled the analysis for potential confounding factors. Population column should report also on the gender of included individuals.

Detailed scoring of the GRADE assessment should be provided as supplementary material. Authors should also add MOOSE or PRISMA checklist.

Author Response

Response to reviewer 3, round 1

The main limitation of the study is that the search is limited to publication date, thus the study cannot provide an evidence-based information. This should be listed among the limitations of the study.

  • Author’s response: Thank you very much for reviewing our manuscript and for your valuable comments for improvement. Unfortunately, the methodological discussion (line 368-399), was missing in the previous version of the paper which we are very sorry for. We have taken all reviewers comments in consideration and all changes in the manuscript is written in red. As stated on line 67-70, our study is contributing to knowledge after Berk 2012. Berk covered studies published until august 2012, our study covers from august 2019.

Line 92 it is necessary to mention which tool was used in order to asses study quality and provide adequate reference.

  • Author’s response: Thank you very much. This is stated on line 102-105 with a reference. We added quality to make sure that this is clear.

Description of inclusion/exclusion criteria should be improved (example information on the eligible study design is lacking).

  • Author’s response: Thank you. We have added publication date for inclusion on line 84, removed dates as an exclusion criteria and inserted eligible study design.

Authors should provide all search terms used in the databases.

  • Author’s response: Thank you for this comment and this is now clarified on line 91.

I have serious concern regarding search criteria as several studies that are fitting inclusion criteria were omitted. Authors should implement the following studies investigating the association between dietary polyphenols intake and depressive symptoms in the systematic review: PMID: 27413131, PMID: 29695122.

  • Author’s response: Thank you very much. We have included the suggested articles in this review (Table 2, and on line 226-233).

Table 1 Information in column “aim” should be provided as two separates columns “exposure/intervention”, “outcomes”. Moreover, authors should provide information whether results reported in the eligible studies controlled the analysis for potential confounding factors. Population column should report also on the gender of included individuals.

  • Author’s response: Thank you for your comment. “Exposure/intervention”, “outcomes” line 82. Confounding factors for the studies is presented in the result section on line 139-141. Regarding gender, adults are established as a person who is fully grown or developed, man or woman. This is stated in Table 2. When it´s only men or woman this is noted, as well as age.

Detailed scoring of the GRADE assessment should be provided as supplementary material. Authors should also add MOOSE or PRISMA checklist.

  • Author’s response: Thank you and since very few RCT studies were found, a GRADE assessment didn´t seem valuable for this study. PRISMA is now provided as supplementary material.

Round 2

Reviewer 1 Report

I appreciate the answers to the questions.

I read the article carefully and overall I liked the changes introduced.

It was a pleasure to read the final version of the article.

Reviewer 3 Report

Thank you for addressing all the comments. Article is suitable for publication.